# On Infinite Separations Between Simple and Optimal Mechanisms

**Alexandros Psomas** [*]    **Ariel Schvartzman** [†]    **S. Matthew Weinberg** [‡]

## Abstract

We consider a revenue-maximizing seller with $k$ heterogeneous items for sale to a single additive buyer, whose values are drawn from a known, possibly correlated prior $\mathcal{D}$. It is known that there exist priors $\mathcal{D}$ such that simple mechanisms — those with bounded menu complexity — extract an arbitrarily small fraction of the optimal revenue ([BCKW15, HN19]). This paper considers the opposite direction: given a correlated distribution $\mathcal{D}$ witnessing an infinite separation between simple and optimal mechanisms, what can be said about $\mathcal{D}$?

[HN19] provides a framework for constructing such $\mathcal{D}$: it takes as input a sequence of $k$-dimensional vectors satisfying some geometric property, and produces a $\mathcal{D}$ witnessing an infinite gap. Our first main result establishes that this framework is without loss: *every* $\mathcal{D}$ witnessing an infinite separation could have resulted from this framework. An earlier version of their work provided a more streamlined framework [HN13]. Our second main result establishes that this restrictive framework is *not* tight. That is, we provide an instance $\mathcal{D}$ witnessing an infinite gap, but which provably could not have resulted from the restrictive framework.

As a corollary, we discover a new kind of mechanism which can witness these infinite separations on instances where the previous "aligned" mechanisms do not.

## 1 Introduction

Consider a revenue-maximizing seller with $k$ items for sale to a single additive buyer, whose values for the $k$ items are drawn from a known distribution $\mathcal{D}$. When $k = 1$, Myerson's seminal work provides a closed-form solution to the revenue-optimal mechanism, and it has a particularly simple form: simply post a price $p := \arg\max_p\{p \cdot \Pr_{v \leftarrow \mathcal{D}}[v \geq p]\}$, and let the buyer purchase the item if they please ([Mye81]). For $k > 1$, however, this *multi-dimensional mechanism design* problem remains an active research agenda forty years later.

While simple, constant-factor approximations are known in quite general settings when $\mathcal{D}$ is a product distribution ([CHK07, CHMS10, CMS15, KW12, LY13, BILW14, Yao15, RW15, CDW16, CM16, CZ17]), there may be an *infinite* gap between the revenue-optimal auction and any simple counterpart when values are correlated ([BCKW15, HN19]). More specifically: one simple way to sell $k$ items is to treat the grand bundle of all items as if it were a single item, and sell it using Myerson's optimal

---

[*]Department of Computer Science, Purdue University, apsomas@cs.purdue.edu. Supported in part by an NSF CAREER award CCF-2144208, a Google Research Scholar Award, and by the Algorand Centres of Excellence program managed by Algorand Foundation. Any opinions, findings, and conclusions or recommendations expressed in this material are those of the author(s) and do not necessarily reflect the views of Algorand Foundation.

[†]Center for Discrete Mathematics and Theoretical Computer Science (DIMACS), schvartman.ariel@gmail.com. Research conducted while at DIMACS, Rutgers University and supported by the National Science Foundation under grant number CCF-1445755.

[‡]Department of Computer Science, Princeton University, smweinberg@princeton.edu. Supported by NSF CCF-1717899

auction (which sets price $\arg\max_p \{p \cdot \Pr_{\vec{v} \leftarrow \mathcal{D}}[\sum_i v_i \geq p]\}$). Letting $\mathrm{BRev}(\mathcal{D})$ denote the revenue achieved by this simple scheme, [HN19] further show a connection between $\mathrm{BRev}(\mathcal{D})$ and *any* simple mechanism through the lens of menu complexity: any mechanism with menu complexity at most $C$ generates expected revenue at most $C \cdot \mathrm{BRev}(\mathcal{D})$.

These works establish a strong separation between simple and optimal auctions: *even when $k = 2$, there exist distributions $\mathcal{D}$ such that $\mathrm{Rev}(\mathcal{D}) = \infty$ (the optimal revenue) while $\mathrm{BRev}(\mathcal{D}) = 1$. The fact that $\mathrm{Rev}(\mathcal{D}) = \infty$ does not on its own suggest that $\mathcal{D}$ must be "weird" (the one-dimensional distribution with CDF $1 - 1/\sqrt{x}$ has this property). The "weird" property is that $\mathrm{Rev}(\mathcal{D})/\mathrm{BRev}(\mathcal{D}) = \infty$, which can never occur for a one-dimensional distribution.

[BCKW15] and [HN19] establish sufficient conditions for a distribution $\mathcal{D}$ to satisfy $\mathrm{Rev}(\mathcal{D})/\mathrm{BRev}(\mathcal{D}) = \infty$. Simply put, the goal of this paper is to study *necessary* conditions for a distribution to satisfy $\mathrm{Rev}(\mathcal{D})/\mathrm{BRev}(\mathcal{D}) = \infty$. We provide two main results. The first establishes that the sufficient condition presented in [HN19] *is in fact necessary* for $\mathrm{Rev}(\mathcal{D})/\mathrm{BRev}(\mathcal{D}) = \infty$ (Theorem 6). The second establishes that the sufficient condition used in an earlier version of that work [HN13] and [BCKW15] is *not necessary* (Theorem 9). In establishing Theorem 9, we also construct a distribution $\mathcal{D}$ such that $\mathrm{Rev}(\mathcal{D})/\mathrm{BRev}(\mathcal{D}) = \infty$ yet the mechanism witnessing this provably falls outside the scope of any previous constructions (Corollary 11).

A recent line of work has borrowed tools from machine learning (more specifically, recommender systems) to approximate buyers with complex, high-dimensional valuations via low-dimensional topic models [CD22]. Their work recognizes some settings where finding approximately optimal mechanisms for these buyers reduces to finding approximately optimal mechanisms for buyers whose valuations are approximated by the topic models. In this context, our work helps clarify the limits where this modeling approach is effective: a model is tractable if and only if it steers clear of the [HN19] framework. Beyond this specific direction, we note that there is a significant community at the intersection of machine learning and multi-item auction design, and understanding the limits of tractability is important for this community.

Proper context and formal statements of our results require precise definitions, which we provide in Section 2 immediately below. Section 3 provides formal statements of our results, along with context alongside related work. Subsequent sections provide proofs.

## 2 Preliminaries

We consider an auction design setting with a single buyer, single seller, and $k$ heterogeneous items. Note that our positive results hold for arbitrary $k$, while our constructions use only $k = 2$ (and $k = 1$ is not possible). We use $\mathcal{D}$ to denote a distribution over $\mathbb{R}_{\geq 0}^k$, the (possibly correlated) distribution over the buyer's values for the $k$ items. The buyer is additive, meaning that their value for a set of items $S$ is equal to $\sum_{i \in S} v_i$. We use $\mathrm{Rev}(\mathcal{D})$ to denote the optimal expected revenue achievable by any incentive-compatible mechanism (formally, the supremum of expected revenues, or $\infty$ if the supremum is undefined), and let $\mathrm{BRev}(\mathcal{D})$ denote the revenue achieved by selling the grand bundle as a single item using Myerson's auction.[4] Finally, a *mechanism* $M$ is a set $\{(\vec{q}_i, p_i)\}_i$, where each $\vec{q}_i \in [0,1]^k$ denotes a vector of probabilities, and $p_i \in \mathbb{R}$ denotes a price. When the buyer's valuation is $\vec{v}$, they pay the auctioneer $p_{i(\vec{v})}$, where $i(\vec{v}) := \arg\max_i\{\vec{v} \cdot \vec{q}_i - p_i\}$.[5] $\mathrm{Rev}(\mathcal{D}, M)$ denotes the expected revenue achieved by a particular mechanism $M$ on distribution $\mathcal{D}$. We will also use the shorthand $\vec{q}^M(\vec{v})$ to denote the allocation vector purchased by a vector $\vec{v}$, and $p^M(\vec{v})$ to denote the price paid (we may drop the superscript of $M$ if the mechanism is clear from context).

**Brief Overview of [HN19].** Below, we formally define two geometric properties of sequences of points, which are the focus of this paper. Many of the ideas below appear in both [BCKW15] and [HN19], but we will use the formal definitions from [HN19] (and an earlier published ver-

---

[4]We briefly remind the reader that $\mathrm{BRev}(\mathcal{D})$ serves as a proxy for the achievable revenue by any simple mechanism, especially when focusing on the gap between infinite and finite. For example, the revenue achieved by selling the items separately is at most $k\mathrm{BRev}(\mathcal{D})$, the revenue achieved by any deterministic mechanism is at most $2^k\mathrm{BRev}(\mathcal{D})$, and, more generally, the revenue achieved by any mechanism which offers at most $m$ distinct options is at most $m\mathrm{BRev}(\mathcal{D})$ [HN19].

[5]How ties are broken is irrelevant to our results — all results hold for arbitrary tie-breaking. Also, all mechanisms include an all-zero pair $\vec{q}_0 = (0, \ldots, 0)$, $p_0 = 0$ to insure individual rationality.

sion [HN13]). Below, morally the vectors $\vec{x}_i$ correspond to possible (scaled) valuation vectors, and the vectors $\vec{q}_i$ correspond to possible vectors of allocation probabilities.

**Definition 1.** *Let $X = (\vec{x}_i)_{i=1}^N$ be an ordered sequence of $N$ points ($N$ may be finite, or equal to $+\infty$), with each $\vec{x}_i \in \mathbb{R}_{\geq 0}^k$. Let $Q = (\vec{q}_i)_{i=0}^N$ be another ordered sequence of $N$ points, with each $\vec{q}_i \in [0,1]^k$, and starting with $\vec{q}_0 = (0, ..., 0)$. Define the following:*

$$\mathrm{gap}_i^{X,Q} := \min_{0 \leq j < i} (\vec{q}_i - \vec{q}_j) \cdot \vec{x}_i \qquad \mathrm{MenuGap}(X, Q) := \sum_{i=1}^N \frac{\mathrm{gap}_i^{X,Q}}{||\vec{x}_i||_1}.$$

*We will also slightly abuse notation and define $\mathrm{MenuGap}(X) := \sup_Q \{\mathrm{MenuGap}(X, Q)\}$.*[6]

$\mathrm{MenuGap}(X)$ is some measure of how distinct the angles of points in $X$ are. To get intuition for this, one might try to write a short proof that when $k = 1$, $\mathrm{MenuGap}(X) = 1$ for all $X$ (or that $\mathrm{MenuGap}(X) = 1$ whenever all $\vec{x}_i \in X$ are parallel). We provide such a proof in Appendix A. $\mathrm{MenuGap}(X, Q)$ is just some geometric measure with no obvious intuition for why this quantity should be of interest to auction designers. However, one key result of [HN19] shows that this quantity has connections to simplicity vs. optimality gaps. Specifically, they show:

**Theorem 2** ([HN19], Proposition 7.1). *For every pair of sequences $X = (\vec{x}_i)_{i=1}^N, Q = (\vec{q}_i)_{i=0}^N$ starting with $\vec{q}_0 = (0, ..., 0)$, and all $\varepsilon > 0$, there exists a distribution $\mathcal{D}$ and mechanism $M$ s.t.:*

$$\frac{\mathrm{Rev}(\mathcal{D}, M)}{\mathrm{BRev}(\mathcal{D})} \geq (1 - \varepsilon) \cdot \mathrm{MenuGap}(X, Q).$$

*Moreover, for all $i \in [N]$, the support of $\mathcal{D}$ contains a single point of the form $c_i \vec{x}_i$, for some $c_i \in \mathbb{R}_+$ (and no other points). Additionally, $\vec{q}^M(c_i \vec{x}_i) = \vec{q}_i$.*

The "Moreover,..." portion of Theorem 2 gives some insight to their construction. Further insight can be deduced by observing that the constraint "$\mathrm{gap}_i^{X,Q} \leq \vec{x}_i \cdot (\vec{q}_i - \vec{q}_j)$" looks similar (but far from identical) to an incentive compatibility constraint involving a valuation vector $\vec{x}_i$ and two allocation vectors $\vec{q}_i, \vec{q}_j$. We refer the reader to [HN19] for further details and intuition for this connection. Theorem 2 gives a framework for proving simplicity vs. optimality gaps, but leaves open the question of actually finding a pair of sequences $X, Q$. They approach this through the following observation:

**Definition 3.** *Given a sequence of points $X = (\vec{x}_i)_{i=0}^N \in [0,1]^k$ starting with $\vec{x}_0 = (0, ..., 0)$, define $\mathrm{SupGap}(X)$ (read "support gap of $X$") as $\mathrm{SupGap}(X) := \mathrm{MenuGap}(X, X)$.*

**Observation 4.** *For all $X$, $\mathrm{MenuGap}(X) \geq \mathrm{SupGap}(X)$.*

Finally, [HN13] propose an explicit construction of a sequence $X$ with infinite SupGap.

**Theorem 5** ([HN13]). *There exists an infinite sequence $X$ of points in $[0,1]^2$ s.t. $\mathrm{SupGap}(X) = \infty$.*

Theorems 2 and 5 together yield a two-dimensional distribution with $\mathrm{Rev}(\mathcal{D})/\mathrm{BRev}(\mathcal{D}) = \infty$. It is also worth noting that all prior constructions follow this approach as well. For example, [BCKW15] provides an infinite sequence $X$ of points in $[0,1]^3$ such that $\mathrm{SupGap}(X) = \infty$.

## 3 Our Results

Independent of Observation 4 and Theorem 5, Theorem 2 *alone* provides a framework for constructing distributions $\mathcal{D}$ so that $\mathrm{Rev}(\mathcal{D})/\mathrm{BRev}(\mathcal{D})$ is large: find sequences $X$ so that $\mathrm{MenuGap}(X)$ is large. Our goal is to understand to what extent this framework is *complete* for constructing such instances. Our first main result establishes that *any distribution with $\mathrm{Rev}(\mathcal{D})/\mathrm{BRev}(\mathcal{D}) = \infty$ could have resulted from the framework induced by Theorem 2*. Specifically:

---

[6] Observe that $\mathrm{gap}_i^{X,Q}$ and $\mathrm{MenuGap}(X, Q)$ might be negative. Any claims made throughout this paper regarding $\mathrm{MenuGap}(X, Q)$ are vacuously true when $\mathrm{MenuGap}(X, Q) < 0$. We allow $\mathrm{gap}_i^{X,Q}$ to be negative to match the definition of [HN19] verbatim (although our work will also show that this peculiarity of their definition is not significant).

**Theorem 6.** *For any distribution $\mathcal{D}$ over $k$ items, there exists a sequence of $N$ points ($N$ can be finite, or equal to $+\infty$) $X = (\vec{x}_i)_{i=1}^N$, with each $\vec{x}_i \in supp(\mathcal{D})$, such that*

$$\text{MenuGap}(X) \geq \frac{\text{Rev}(\mathcal{D})}{9\text{BRev}(\mathcal{D})}.$$

*In particular, if $\text{Rev}(\mathcal{D})/\text{BRev}(\mathcal{D}) = \infty$, then $\text{MenuGap}(X) = \infty$ as well.*

A complete proof of Theorem 6 appears in Section 4. Observe also that because $\text{MenuGap}(X)$ is monotone (in the sense that adding points to $X$, anywhere, cannot possibly decrease $\text{MenuGap}(X)$), the fact that $X$ is a subset of the support of $\mathcal{D}$ (rather than the entire support) is immaterial.[7] Put another way, the important aspect in constructing $X$ is how elements in the support of $\mathcal{D}$ are ordered, rather than which points are included.

Observation 4 further provides a framework to construct sequences so that $\text{MenuGap}(X)$ is large: construct sequences $X$ so that $\text{SupGap}(X)$ is large. One may then wonder if $\text{SupGap}(X)$ and $\text{MenuGap}(X)$ are approximately related, for all $X$. For this specific question, the answer is trivially no, due to incompatibility with scaling (multiplying every point in $X$ by $1/2$ will decrease $\text{SupGap}(X)$ by a factor of 2, but not $\text{MenuGap}(X)$). Therefore, not much insight is gained by studying this precise question.

Instead, we observe that the interesting aspect of constructions resulting through $\text{SupGap}(X)$ is that $\vec{x}_i$ and $\vec{q}_i$ are *aligned* (that is, $\vec{x}_i = c_i \cdot \vec{q}_i$ for some $c_i \in \mathbb{R}_{\geq 0}$). Specifically, even if $\vec{q}_i = \vec{x}_i$, this equality is not maintained by the construction of Theorem 2. However, if $\vec{q}_i$ and $\vec{x}_i$ are aligned, this alignment property *is* maintained by the construction. We therefore propose the following definition, which captures the maximum value achievable by $\text{MenuGap}(X, Q)$ when $X, Q$ are aligned.

**Definition 7.** *Let $X = (\vec{x}_i)_{i=1}^N$ be an ordered sequence of $N$ points in $[0,1]^k$ ($N$ may be finite, or equal to $+\infty$). Let also $C = (c_i)_{i=0}^N$ be an ordered sequence of numbers, with each $c_i \in [0, 1/||\vec{x}_i||_\infty]$, starting with $c_0 = 0$. Define:*

$$\text{sgap}_i^{X,C} := \min_{j<i} \vec{x}_i \cdot (c_i \vec{x}_i - c_j \vec{x}_j), \qquad \text{and} \qquad \text{AlignGap}(X, C) := \sum_{i=1}^N \frac{\max\{0, \text{sgap}_i^{X,C}\}}{||\vec{x}_i||_1}.$$

*We will also slightly abuse notation and denote by $\text{AlignGap}(X) := \sup_C \{\text{AlignGap}(X, C)\}$.*

Recall that we have chosen to let $c_i$ range in $[0, 1/||\vec{x}_i||_\infty]$ (rather than be fixed at 1, or $1/||\vec{x}_i||_\infty$) to give potential constructions flexibility in scaling $\vec{q}_i$. Additionally, by ensuring that the contribution of each $\text{sgap}_i^{X,C}$ is non-negative, we give potential constructions flexibility to ignore points in the sequence. That is, any construction using MenuGap directly can always set $\vec{q}_i := \arg\max_{j<i}\{\vec{q}_j \cdot \vec{x}_i\}$, which effectively just drops $\vec{x}_i$ from the sequence. Counting $\max\{0, \text{sgap}_i^{X,C}\}$ towards the objective (rather than just $\text{sgap}_i^{X,C}$) gives constructions that arise through AlignGap the same flexibility.

**Lemma 8.** *For all $X$, $\text{AlignGap}(X) \leq \text{MenuGap}(X)$.*

The proof of Lemma 8 is in Appendix B. Lemma 8 induces a framework to design sequences with large MenuGap: design sequences with large AlignGap. Our second main result establishes that this framework *is not* without loss of generality, even for $k = 2$. Specifically:

**Theorem 9.** *There exist sequences $X = (\vec{x}_i)_{i=1}^\infty \in [0,1]^2$ such that:*

$$\text{AlignGap}(X) \leq 6 \qquad but \qquad \text{MenuGap}(X) = \infty.$$

A complete proof of Theorem 9 appears in Section 5. By the discussion following Definition 7, the source of this gap is entirely due to the requirement that the sequence $Q$ be aligned with $X$ (and is not due to inability to scale, or inability to ignore difficult points in $X$). We make this crisp with the following definition and corollary, which construct a novel distribution witnessing $\text{Rev}(D)/\text{BRev}(D) = \infty$ that is provably distinct from all previous approaches.

**Definition 10.** *For a distribution $\mathcal{D}$ and mechanism $M$, define the* Aligned Revenue *of $M$ on $\mathcal{D}$:*

$$\text{ARev}(\mathcal{D}, M) := \mathbb{E}_{\vec{v} \leftarrow \mathcal{D}}[p^M(\vec{v}) \cdot I(\vec{v} \text{ is parallel to } \vec{q}^M(\vec{v}))], \qquad \text{ARev}(\mathcal{D}) := \sup_M \{\text{ARev}(\mathcal{D}, M)\}.$$

---

[7]Of course, if the support of $\mathcal{D}$ is uncountable, then clearly the entire support of $\mathcal{D}$ cannot be included in $X$.

**Corollary 11.** *There exist distributions $\mathcal{D}$ over two items such that* $\mathrm{Rev}(\mathcal{D})/\mathrm{ARev}(\mathcal{D}) = \infty$.

A proof of Corollary 11 appears in Appendix C. It is worth noting that *all* previous constructions establishing $\mathrm{Rev}(\mathcal{D})/\mathrm{BRev}(\mathcal{D}) = \infty$ proceeded by producing an $X$ such that $\mathrm{SupGap}(X) = \infty$. Indeed, the [BCKW15] construction provides such an $X$ when $k = 3$, the [HN13] construction provides an $X$ when $k = 2$, and the [PSW19] construction adapts parameters in that of [HN13]. By Theorem 2, this implies not only that $\mathrm{Rev}(\mathcal{D})/\mathrm{BRev}(\mathcal{D}) = \infty$, but also that $\mathrm{ARev}(\mathcal{D})/\mathrm{BRev}(\mathcal{D}) = \infty$. Corollary 11 establishes the existence of a fundamentally different construction,[8] as our $\mathcal{D}$ has a finite ratio between $\mathrm{ARev}(\mathcal{D})/\mathrm{BRev}(\mathcal{D})$, yet still maintains an infinite ratio between $\mathrm{Rev}(\mathcal{D})/\mathrm{BRev}(\mathcal{D})$.

**Additional Related Work.** We've already discussed the most related work to ours, which is that of [HN19, BCKW15]. There is also a large body of work studying *product distributions* specifically, and establishes that simple mechanisms can achieve constant factor approximations in quite general settings [CHK07, CHMS10, CMS15, KW12, LY13, BILW14, Yao15, RW15, CDW16, CM16, CZ17]). Recent works have made progress in obtaining arbitrary approximations ([BGN17, KMS$^+$19]), which again rely on the assumption that $\mathcal{D}$ is a product distribution.

Three recent lines of work address the [BCKW15, HN19] constructions in a different manner. First, [CTT19, CTT20] consider the related *buy-many* model (where the auctioneer cannot prevent the buyer from interacting multiple times with the auction). [CTT19] establishes that selling separately achieves an $O(\log k)$-approximation to the optimal buy-many mechanism in *quite* general settings (including the settings considered in this paper). In a different direction, [PSW19] uses the lens of smoothed analysis ([ST04]) to reason about the robustness of the [HN19] constructions. Finally, [Car17] considers a correlation robust framework in which the valuation profile is drawn from a correlated distribution that is not completely known to the seller; the goal is to design a mechanism that maximizes the worst-case (over correlations) seller revenue, when only the items' marginal distributions are known. [Car17] shows that selling each item separately is optimal; see [BGLT19, GL18] for further work in this model.

## 4 A Converse to Theorem 2: The [HN19] Framework is WLOG

In this section, we prove Theorem 6. Our proof has two main parts. First, we will take the optimal auction for $\mathcal{D}$ (or one that is arbitrarily close to optimal) and repeatedly simplify it through a sequence of lemmas, at the cost of small constant-factors of revenue. The second part takes this simple menu and draws a connection to MenuGap. Missing proofs can be found in Appendix B.

**Simplifying the Optimal Mechanism.** We show that for every $\mathcal{D}$, there exists an approximately-optimal mechanism which satisfies some useful properties. First, we argue that we may ignore menu options with low prices.

**Definition 12.** *We say that a mechanism $M$ is $c$-expensive if every option has price at least $c$.*

**Claim 13.** *For all $c \in \mathbb{R}_{\geq 0}$, all distributions $\mathcal{D}$, and all mechanisms $M$, there exists a $c$-expensive mechanism $M'$ satisfying* $\mathrm{Rev}(\mathcal{D}, M') \geq \mathrm{Rev}(\mathcal{D}, M) - c$.

Our next step will show that we can assume further structure on the prices charged, at the cost of a factor of two.

**Definition 14.** *A $c$-expensive mechanism $M$ is* oddly-priced *(respectively,* evenly-priced*) if for all $\vec{v}$, there exists an odd (respectively, even) integer $i$ such that $p^M(\vec{v}) \in [c \cdot 2^i, c \cdot 2^{i+1})$.*

**Claim 15.** *For all $c$-expensive mechanisms $M$, and all $\mathcal{D}$, there exists either an oddly-priced or evenly-priced $c$-expensive mechanism $M'$ satisfying* $\mathrm{Rev}(\mathcal{D}, M') \geq \mathrm{Rev}(\mathcal{D}, M)/2$.

This concludes our simplification of the mechanism. In the subsequent section, we draw a connection between MenuGap and the revenue of oddly- or evenly-price $c$-expensive mechanisms.

**Connecting Structured Mechanisms to MenuGap.** We begin with the following definition, which describes our proposed $X, Q$ based on a structured mechanism for $\mathcal{D}$.

---

[8]On a technical level, our construction certainly borrows several ideas from previous ones, however.

**Definition 16** (Representative Sequences). *Let $M$ be a $c$-expensive mechanism which is oddly-priced or evenly-priced, and let $\mathcal{D}$ be any distribution. An $\varepsilon$-representative sequence for $M, \mathcal{D}$ is the following:*

- *Define $\vec{q}_0(\mathcal{D}, M) := (0, \ldots, 0)$.*

- *Define offset $a$ to be $1$ if $M$ is oddly-priced, and $0$ if $M$ is evenly-priced.*

- *For all $j \in \mathbb{N}_+$, define $B_j := \{\vec{v} \in supp(\mathcal{D}) : p^M(\vec{v}) \in [c \cdot 2^{2(j-1)+a}, c \cdot 2^{2(j-1)+a+1})\}$.*

- *For all $j \in \mathbb{N}_+$, let $\vec{x}_j \in B_j$ be such that $||\vec{x}_j||_1 \leq ||\vec{v}||_1 \cdot (1+\varepsilon)$ for all $\vec{v} \in B_j$.[9]*

- *For all $j \in \mathbb{N}_+$, let $\vec{q}_j := \vec{q}^M(\vec{x}_j)$.*

**Proposition 17.** *Let $M$ be a $c$-expensive, and oddly- or evenly-priced. Let $(X, Q)$ be an $\varepsilon$-representative sequence for $M, \mathcal{D}$. Then: $\mathrm{MenuGap}(X, Q) \geq \frac{\mathrm{Rev}(\mathcal{D}, M)}{4(1+\varepsilon)\mathrm{BRev}(\mathcal{D})}$.*

*Proof.* The proof will follow from two technical claims. The first claim relates $||\vec{x}_i||_1$ and $\mathrm{BRev}(\mathcal{D})$. The second claim relates $\mathrm{Rev}(\mathcal{D}, M)$ and $\mathrm{MenuGap}(X, Q)$. Crucially, this claim uses the fact that the mechanism is either oddly-priced or evenly-priced (and therefore $p^M(\vec{x}_i), p^M(\vec{x}_j)$ differ by at least a factor of $2$, for any $i \neq j$).

**Claim 18.** $(1 + \varepsilon) \cdot \mathrm{BRev}(\mathcal{D}) \geq ||\vec{x}_i||_1 \cdot \Pr_{\vec{v} \sim \mathcal{D}}[\vec{v} \in B_i]$.

**Claim 19.** *If $X, Q$ is an $\varepsilon$-representative sequence for $M, \mathcal{D}$, then $\mathrm{gap}_i^{X,Q} \geq p^M(\vec{x}_i)/2$.*

With these two claims, we can complete the proof of Proposition 17:

$$\mathrm{MenuGap}(X, Q) := \sum_{i=1}^{\infty} \frac{\mathrm{gap}_i^{X,Q}}{||\vec{x}_i||_1}$$

$$\geq^{(Claim\ 19)} \sum_{i=1}^{\infty} \frac{p^M(\vec{x}_i)}{2 \cdot ||\vec{x}_i||_1}$$

$$\geq^{(Claim\ 18)} \sum_{i=1}^{\infty} \frac{p^M(\vec{x}_i) \cdot \Pr_{\vec{v} \sim \mathcal{D}}[\vec{v} \in B_i]}{2 \cdot (1+\varepsilon)\mathrm{BRev}(\mathcal{D})}$$

$$\geq \sum_{i=1}^{\infty} \frac{\mathbb{E}_{\vec{v} \sim \mathcal{D}|\vec{v} \in B_i}[p^M(\vec{v})] \cdot \Pr_{\vec{v} \sim \mathcal{D}}[\vec{v} \in B_i]}{4 \cdot (1+\varepsilon)\mathrm{BRev}(\mathcal{D})}$$

$$\geq \frac{\mathrm{Rev}(\mathcal{D}, M)}{4 \cdot (1+\varepsilon)\mathrm{BRev}(\mathcal{D})}.$$

Above, the first line is the definition of MenuGap. The fourth line uses the fact that for all $\vec{v} \in B_i$, $p^M(\vec{v}) \leq 2^{2(i-1)+a+1} = 2 \cdot 2^{2(i-1)+a} \leq 2p^M(\vec{x}_i)$. The final line is just rewriting the definition of $\mathrm{Rev}(\mathcal{D}, M)$, and using the fact that every $\vec{v} \in \mathrm{supp}(\mathcal{D})$ with $p(\vec{v}) > 0$ is in some $B_i$. $\square$

**Wrapping Up.** Proposition 17, together with the fact that an oddly- or evenly-priced mechanism is guaranteed to get a good approximation to the optimum, now suffices to prove Theorem 6.

*Proof of Theorem 6.* Set $c = \mathrm{Rev}(\mathcal{D})/100$, and $\varepsilon = 1/100$. Note that $\varepsilon$-representative sequences are guaranteed to exist for any $M, \mathcal{D}$, as $\varepsilon > 0$. Let $M'$ denote the $c$-expensive mechanism promised by Claim 13. Let then $M''$ denote the oddly- or evenly-priced mechanism promised by Claim 15. Finally, let $X, Q$ denote the $\varepsilon$-representative sequence for $M'', \mathcal{D}$. We get:

$$\mathrm{MenuGap}(X, Q) \geq^{(\mathrm{Proposition\ 17})} \frac{\mathrm{Rev}(\mathcal{D}, M'')}{4(1+\varepsilon)\mathrm{BRev}(\mathcal{D})}$$

---

[9]Note that for all $\varepsilon > 0$, such an $\vec{x}_j$ exists, even if $B_j$ is not closed (as long as $B_j$ is non-empty). In particular, because $M$ is $c$-expensive, we know that $||\vec{v}||_1 \geq c > 0$ for all $\vec{v}$ who pay $\geq c$. If $B_j$ is empty, instead omit $\vec{x}_j, \vec{q}_j$ from both lists (i.e. decrease all future indices by one).

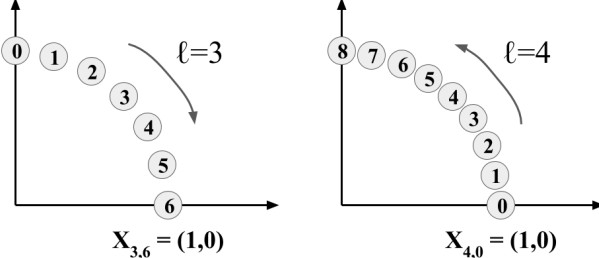

Figure 1: An illustration of two layers of our construction. The number of points in each layer increases with $\ell$, but they are always evenly spaced between $(1,0)$ and $(0,1)$. The direction in which the points are placed alternates between clockwise and counterclockwise.

$$\geq^{\text{(Claim 15)}} \frac{\mathrm{Rev}(\mathcal{D}, M')}{8(1+\varepsilon)\mathrm{BRev}(\mathcal{D})}$$

$$\geq^{\text{(Claim 13)}} \frac{\mathrm{Rev}(\mathcal{D}) - c}{8(1+\varepsilon)\mathrm{BRev}(\mathcal{D})}$$

$$\geq \frac{\mathrm{Rev}(\mathcal{D})}{9\mathrm{BRev}(\mathcal{D})}. \qquad\qquad \square$$

## 5  No Converse to Lemma 8: Separating MenuGap and AlignGap

In this section we prove our second main result: Lemma 8 does *not* admit a converse, even approximately. We briefly remind the reader that *all* previous constructions witnessing $\mathrm{Rev}(\mathcal{D})/\mathrm{BRev}(\mathcal{D}) = \infty$ arose by establishing sequences $X$ with $\mathrm{AlignGap}(X) = \infty$ (in fact, even $\mathrm{SupGap}(X) = \infty$). Theorem 9 establishes that constructions exist outside of this more restrictive framework. Missing proofs can be found in Appendix B.

**Description of our construction.**  Our infinite sequence $X$ consists of consecutive *layers* of points. For $\ell = 2$ to $\infty$, layer $\ell$ has $n_\ell := \ell\lceil\ln^2(\ell)\rceil + 1$ points. These points/vectors have $\ell_2$ norm equal to one, and are evenly spaced (in terms of their angle) between $(1,0)$ and $(0,1)$. If $\ell$ is even, they go counterclockwise from $(1,0)$ to $(0,1)$. If $\ell$ is odd, they go clockwise from $(0,1)$ to $(1,0)$:

- Define $n_\ell := \ell\lceil\ln^2(\ell)\rceil + 1$. Define $\theta_\ell := \frac{\pi}{2(n_\ell-1)}$.

- Point $\vec{x}_{\ell,j}$ is the $j^{th}$ point in the $\ell^{th}$ layer, and is $(\cos(j\theta_\ell), \sin(j\theta_\ell))$ when $\ell$ is even, or $(\sin(j\theta_\ell), \cos(j\theta_\ell))$ when $\ell$ is odd.

- The infinite sequence $X$ is $((\vec{x}_{\ell,j})_{j=0}^{n_\ell-1})_{\ell=2}^{\infty}$.

Figure 1 demonstrates two layers of our construction. In the remainder of this section, we may refer to the sequence of points $\vec{x}_{\ell,j}$ by a single indexed sequence $\vec{x}_i$. The latter is the same as the former where points are ordered lexicographically with respect to the original indexing.

**Upper Bounding** $\mathrm{AlignGap}(X)$ **via Lagrangian duality.**  We first upper bound $\mathrm{AlignGap}(X)$ and establish that it's finite. To this end, observe that for any sequence $X$, $\mathrm{AlignGap}(X)$ is the solution to the following (infinite, if $X$ is infinite) mathematical program, where the variables are $\mathrm{sgap}_i, c_i$ (the sequence $\vec{x}_i$ is fixed, as we're aiming to compute $\mathrm{AlignGap}(X)$):

$$\mathrm{AlignGap}(X): \begin{cases} & \max \sum_i \max\{0, \mathrm{sgap}_i\}/||\vec{x}_i||_1 \\ \text{subject to:} & \forall i, j < i : \mathrm{sgap}_i \leq \vec{x}_i \cdot (c_i\vec{x}_i - c_j\vec{x}_j) \\ & \forall i : 0 \leq c_i \leq 1/||\vec{x}_i||_\infty \end{cases}$$

We next proceed with a series of relaxations of this program. Some steps are specific to our choice of $X$ from Section 5, while others hold for arbitrary $X$. Our first step is specific to this construction, and simply bounds $||\vec{x}||_1$. Consider the following mathematical program:

$$\text{AlignGap}'(X) : \begin{cases} \max \sum_i \max\{0, \text{sgap}_i\} \\ \text{subject to:} \quad \forall i, j < i : \text{sgap}_i \leq \vec{x}_i \cdot (c_i \vec{x}_i - c_j \vec{x}_j) \\ \forall i : 0 \leq c_i \leq \sqrt{2}. \end{cases}$$

**Observation 20.** *For the sequence $X$ defined in Section 5,* $\text{AlignGap}'(X) \geq \text{AlignGap}(X)$.

*Proof.* Every $\vec{x}_i$ in the construction has $||\vec{x}_i||_1 \geq ||\vec{x}_i||_2 = 1$. Therefore, the new objective function is only larger. Moreover, every $\vec{x}_i$ in the construction has $||\vec{x}_i||_\infty \geq 1/\sqrt{2}$ (because the $\ell_2$ norm is 1), so this is relaxing the upper bound on $c_i$. $\qquad\square$

We proceed to upper bound $\text{AlignGap}'(X)$ via a Lagrangian relaxation of the formulation above. Consider the following Lagrangian relaxation. We put a Lagrangian multiplier of 1 on every constraint of the form $\text{sgap}_i \leq \vec{x}_i \cdot (c_i \vec{x}_i - c_{i-1} \vec{x}_{i-1})$, for all $i > 1$, and a Lagrangian multiplier of 0 on all other constraints involving sgap. We will *not* put a Lagrangian multiplier on constraints binding $c_i$ to $[0, \sqrt{2}]$, and keep those in the program. This yields the following Lagrangian relaxation (for simplicity of notation below, define $c_0 := 0$, and define $\vec{x}_0 = \vec{0}$):

$$\text{LagRel}_1(X) : \begin{cases} \max \sum_i \max\{0, \text{sgap}_i\} + \vec{x}_i \cdot (c_i \vec{x}_i - c_{i-1} \vec{x}_{i-1}) - \text{sgap}_i \\ \text{subject to:} \quad \forall i : 0 \leq c_i \leq \sqrt{2} \end{cases}$$

**Observation 21.** *For all $X$,* $\text{AlignGap}'(X) \leq \text{LagRel}_1(X)$.

We now proceed to further simplify $\text{LagRel}_1(X)$. The next step is defined below:

$$\text{LagRel}_2(X) : \begin{cases} \max \sum_i \vec{x}_i \cdot (c_i \vec{x}_i - c_{i-1} \vec{x}_{i-1}) \\ \text{subject to:} \quad \forall i : 0 \leq c_i \leq \sqrt{2} \end{cases}$$

**Observation 22.** *For all $X$,* $\text{LagRel}_1(X) = \text{LagRel}_2(X)$.

Finally, we can rewrite the objective function to group all coefficients of $c_i$. For ease of notation below, define $\vec{x}_{N+1} := \vec{0}$ (if $N$ is finite. If $N = \infty$, there are no notational issues).

$$\text{LagRel}(X) : \begin{cases} \max \sum_i c_i \vec{x}_i \cdot (\vec{x}_i - \vec{x}_{i+1}) \\ \text{subject to:} \quad \forall i : 0 \leq c_i \leq \sqrt{2} \end{cases}$$

**Observation 23.** *For all $X$,* $\text{AlignGap}'(X) \leq \text{LagRel}_1(X) = \text{LagRel}_2(X) = \text{LagRel}(X)$.

Now, we move to analyze $\text{LagRel}(X)$ for our particular sequence $X$.

**Claim 24.** *For the sequence $X$ defined in Section 5,* $\text{LagRel}(X) = \sqrt{2} \cdot \sum_i 1 - \vec{x}_i \cdot \vec{x}_{i+1}$.

*Proof.* For all $i$, $\vec{x}_i \cdot (\vec{x}_i - \vec{x}_{i+1}) \geq 0$, as each $\vec{x}_i$ has $\ell_2$ norm exactly one. Thus, the optimal solution for $\text{LagRel}(X)$ sets each $c_i := \sqrt{2}$. Recalling that $\vec{x}_i \cdot \vec{x}_i = 1$ for all $i$ concludes the claim. $\qquad\square$

**Proposition 25.** *For the sequence $X$ defined in Section 5,* $\text{LagRel}(X) \leq 6$.

*Proof.* Let us first observe that if $\vec{x}_i$ is the last point in a layer, then in fact $\vec{x}_{i+1} = \vec{x}_i$, and therefore $1 - \vec{x}_i \cdot \vec{x}_{i+1} = 0$. Therefore, these terms do not contribute to the sum. We can then rewrite the term to sum over all layers as follows:

$$\text{LagRel}(X) = \sqrt{2} \cdot \sum_\ell \sum_{j=0}^{n_\ell - 2} 1 - \vec{x}_{\ell,j} \cdot \vec{x}_{\ell,j+1}$$
$$= \sqrt{2} \cdot \sum_\ell (n_\ell - 1) \cdot (1 - \cos(\theta_\ell))$$

$$\leq \sqrt{2} \cdot \sum_{\ell} (n_\ell - 1) \cdot \theta_\ell^2 / 2$$

$$= \sqrt{2} \cdot \sum_{\ell} \frac{\pi^2}{8(n_\ell - 1)}$$

$$= \sqrt{2} \cdot \sum_{\ell} \frac{\pi^2}{8\ell \lceil \ln^2(\ell) \rceil}$$

$$\leq 6.$$

Above, the first line follows by the reasoning in the first paragraph. The second line follows by observing that the angle between any two points in layer $\ell$ is exactly $\theta_\ell$ (and the two points in question of $\ell_2$ norm equal to one). The third line follows as $\cos(\theta_\ell) \geq 1 - \theta_\ell^2/2$ for any $\theta_\ell \in [0, \pi/2]$ (and all $\theta_\ell$ are indeed in $[0, \pi/2]$). The fourth line follows by substituting the definition of $\theta_\ell$ as a function of $n_\ell$. The fifth line follows by definition of $n_\ell$. The final line is just calculation for this particular infinite series (this follows as $\sum_{\ell=2}^{\infty} \frac{1}{\ell \ln^2(\ell)} \leq 3$).  □

Observation 20, Observation 23, and Proposition 25 yield the main result of this section:

**Proposition 26.** *For the sequence $X$ defined in Section 5,* $\mathrm{AlignGap}(X) \leq 6$.

**Step 3: Picking a $Q$ to Lower Bound** $\mathrm{MenuGap}(X)$. Finally, we propose a sequence $Q$ and show that $\mathrm{MenuGap}(X, Q) = \infty$. We describe the sequence again in layers, to match our description of $X$ (that is, the vector $\vec{q}_{\ell,j}$ corresponds to the vector $\vec{x}_{\ell,j}$). In particular, for each even layer $\ell$, the vectors $\vec{q}_{\ell,j}$ will have a fixed $x$-coordinate, and the $y$-coordinate will increase with $j$. For each odd layer, we will introduce no new vectors (i.e. we will just let $\mathrm{gap}_i^{X,Q} = 0$ for all $i$ in an odd layer). Specifically, the construction is as follows:

- Define $\alpha := \sum_{\ell=2}^{\infty} \frac{1}{\ell \ln^2(\ell)}$ (and note that $\alpha < \infty$).

- Define $z_\ell := \frac{1}{\alpha} \sum_{j=2}^{\ell} \frac{1}{j \ln^2(j)}$.

- Define $\delta_\ell := z_\ell - z_{\ell-1} = \frac{1}{\alpha \ell \ln^2(\ell)}$.

- For $j < n_\ell - 1$, define $z_{\ell,j} := 1 - \delta_\ell \cot((j+1)\theta_\ell)$. For $j = n_\ell - 1$, define $z_{\ell,j} := 1$.[10]

- For all even $\ell$, and all $j$, set $\vec{q}_{\ell,j} := (z_\ell, z_{\ell,j})$.

- For all odd $\ell$, and all $j$, set $\vec{q}_{\ell,j} := \arg\max_{\ell' < \ell, j'} \{\vec{x}_{\ell,j} \cdot \vec{q}_{\ell',j'}\}$.

**Proposition 27.** $\mathrm{MenuGap}(X, Q) = \infty$.

In our proof (Appendix B) we first analyze which point sets the gap for $\vec{x}_{\ell,j}$, and observe that it must either be $\vec{q}_{\ell,j-1}$ or $\vec{q}_{\ell-2,n_{\ell-2}-1}$ (that is, it must be the previous point in the same layer, or the final point in the previous even layer). We can then bound the gap of each layer exactly, and then take a sum over layers: for any even layers we have that the sum of gaps is at least $\sum_{j=0}^{n_\ell - 1} \mathrm{gap}_{\ell,j}^{X,Q} \geq \delta_\ell \cdot \ln(n_\ell)/2$, where $\mathrm{gap}_{\ell,j}^{X,Q} := \mathrm{gap}_i^{X,Q}$, where $\vec{x}_i := \vec{x}_{\ell,j}$ ($\vec{x}_i$ is the $j^{th}$ point on layer $\ell$). Recall that $\delta_\ell := \frac{1}{\alpha n_\ell} = \frac{1}{\alpha \ell \ln^2(\ell)}$ we conclude that:

$$\sum_{\ell \text{ even}} \sum_{j=0}^{n_\ell - 1} \mathrm{gap}_{\ell,j}^{X,Q} \geq \sum_{\ell \text{ even}} \delta_\ell \cdot \ln(n_\ell)/2 = \sum_{\ell \text{ even}} \frac{1}{2\alpha \ell \ln(\ell)} = \infty.$$

This completes the proof of Theorem 9: Proposition 26 establishes that $\mathrm{AlignGap}(X) \leq 6$, while Proposition 27 establishes that $\mathrm{MenuGap}(X) = \infty$.

---

[10]To see that $z_{\ell,j} \geq 0$ observe that $\cot(x) \leq 1/x$ for all $x \in [0, \pi/2]$. Therefore we get $\delta_\ell \cot((j+1)\theta_\ell) \leq \delta_\ell \frac{1}{(j+1)\theta_\ell} \leq \delta_\ell \frac{2(n_\ell - 1)}{\pi(j+1)} \leq \frac{2}{\pi} \delta_\ell n_\ell = \frac{2\ell \lceil \log^2 \ell \rceil}{\pi \alpha \ell \log^2 \ell} \leq 1$, for all $\ell, j$ ($\alpha > 1.9$). To see that $z_{\ell,j} \leq 1$ simply note that the term we subtract can't be negative.

# 6  Conclusion

We study the nature of distributions $\mathcal{D}$ with $\mathrm{Rev}(\mathcal{D})/\mathrm{BRev}(\mathcal{D}) = \infty$. Prior work established a framework to construct such distributions, and therefore established sufficient conditions [BCKW15, HN19]. Our first main result establishes that the most general of these frameworks is in fact complete (Theorem 6). Our second main result establishes that the more restrictive framework, through which all previous constructions arose, is not complete (Theorem 9). Finally, we build upon our main construction to develop a novel distribution $\mathcal{D}$ witnessing $\mathrm{Rev}(\mathcal{D})/\mathrm{BRev}(\mathcal{D}) = \infty$, but for which none of the "aligned" mechanisms of prior work can possibly witness this (Corollary 11).

In terms of future work, it remains open as to whether there is an alternative definition for "sequences with $\mathrm{MenuGap}(X) = \infty$" which is easier to parse. Our work establishes that understanding such sequences is necessary and sufficient to understand distributions with $\mathrm{Rev}(\mathcal{D})/\mathrm{BRev}(\mathcal{D}) = \infty$, and establishes that "sequences with $\mathrm{MenuGap}(X) = \infty$" is not equivalent to "sequences with $\mathrm{AlignGap}(X) = \infty$." But it would be exciting for future work to better understand sequences with $\mathrm{MenuGap}(X) = \infty$.

Interestingly, ongoing work by [AS22] uses the framework developed in this paper in order to prove *approximation results* for so-called fine-grained buy-many mechanisms, a class of mechanisms which interpolates between buy-one and the recently introduced buy-many mechanisms [CTT19, CTT20]. In general, our techniques make it possible for future work to explore the gap between various simple versus optimal benchmarks without having to reason directly about the underlying mechanisms.

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
