# A MenuGap$(X) = 1$ when $k = 1$

In this brief section we prove that when $k = 1$, for any sequence of $x_i \in \mathbb{R}^+_{\geq 0}$, MenuGap$(X) = 1$.

**Claim 28.** *When $k = 1$, for any $X = \{x_i\}_{i=1}^N$, $x_i \in \mathbb{R}^+_{\geq 0}$, MenuGap$(X) = 1$.*

*Proof.* Note that when $k = 1$, $||x_i||_1 = x_i$. Therefore, MenuGap$(X, Q) = \sum_i \min_{j<i}(q_i - q_j)$. We make the following observation which allows us to look at structured optimal solutions.

**Observation 29.** *Any optimal solution $Q$ to MenuGap$(X)$ is monotone non-decreasing.*

*Proof.* For the sake of contradiction, suppose we are given an optimal solution $Q$ that is not monotone non-decreasing. Let $i$ be the smallest index for which $q_i < q_{i-1}$. Then $\text{gap}_i^{X,Q} = (q_i - q_{i-1})x_i < 0$. Consider instead a solution $Q'$ where $q'_j = q_j$ for all $j \neq i$ and $q'_i = q_{i-1}$. Now, $\text{gap}_i^{X,Q'} = 0$. Since $Q_{\leq i-1} = Q'_{\leq i-1}$, $\text{gap}_j^{X,Q} = \text{gap}_j^{X,Q'}$ for all $j < i$. Since $q_{i-1} > q_i$, for any $j > i$ it holds that $(q_j - q_{i-1}) < (q_j - q_i)$. Therefore, $q_i$ is not "setting the gap" for any point after it. Hence it also holds that $\text{gap}_j^{X,Q} = \text{gap}_j^{X,Q'}$ for all $j > i$. Putting everything together we get that MenuGap$(X, Q') - $MenuGap$(X, Q) = \text{gap}_i^{X,Q'} - \text{gap}_i^{X,Q} > 0$ contradicting the optimality of $Q$. $\square$

With this observation in hand, since the $q_i$ are monotone non-decreasing, without loss of generality it holds that $\text{gap}_i^{X,Q} = \min_{j<i} q_i - q_j = q_i - q_{i-1}$ ($q_{i-1} \geq q_j$ for all $j < i$). Therefore, we get MenuGap$(X, Q) = \sum_i q_i - q_{i-1} = q_N - q_0$. Since $q_0 = 0$ and $0 \leq q_N \leq 1$, we get that MenuGap$(X, Q) \leq 1$.

Finally, note that for any $X$, we can set $q_N = 1$ and $q_i = 0$ for all other $i$, proving that MenuGap$(X) \geq 1$.

$\square$

# B Omitted Proofs

*Proof of Lemma 8.* We prove that for all $X, C$, AlignGap$(X, C) \leq $ MenuGap$(X)$, which implies the lemma. For a given $X, C$, define:

- $\vec{q}_i := c_i \cdot \vec{x}_i$, if $\text{sgap}_i^{X,C} > 0$.

- $\vec{q}_i := \arg\max_{j<i}\{c_j \cdot \vec{x}_j\}$, if $\text{sgap}_i^{X,C} \leq 0$.

Observe first that each $\vec{q}_i \in [0, 1]^k$, as each $c_i \vec{x}_i \in [0, 1]^k$ (this follows because each component of $\vec{x}_i$ is at most $||\vec{x}_i||_\infty$, and each $c_i$ is at most $1/||\vec{x}_i||_\infty$). Next, observe that if $\text{sgap}_i^{X,C} \leq 0$, then $\text{gap}_i^{X,Q} = 0$. This is by definition in bullet two above. Finally, observe that if $\text{sgap}_i^{X,C} > 0$, then $\text{gap}_i^{X,Q} \geq \text{sgap}_i^{X,C}$. This is because the set of $\{\vec{q}_j\}_{j<i}$ is a subset of $\{c_j \vec{x}_j\}_{j<i}$, and because $\vec{q}_i := c_i \cdot \vec{x}_i$ by bullet one. Therefore, $\text{gap}_i^{X,Q} \geq \max\{0, \text{sgap}_i^{X,C}\}$ for all $i$ and the lemma follows. $\square$

*Proof of Claim 13.* Take $M'$ to be exactly the same as $M$, except having removed all entries with price $< c$. For every value in the support of $\mathcal{D}$ with $p^M(\vec{v}) \geq c$ in $M$, we still have $p^{M'}(\vec{v}) \geq c$. This is simply $\vec{v}$'s favorite option in $M$ is still available in $M'$, and all options in $M'$ were also available in $M$. For any value with $p^M(\vec{v}) < c$, we clearly have $p^{M'}(\vec{v}) \geq 0$. So for all $\vec{v}$, we have $p^{M'}(\vec{v}) \geq p^M(\vec{v}) - c$, and the claim follows by taking an expectation with respect to $\vec{v}$. $\square$

*Proof of Claim 15.* Simply let $M_1$ denote the set of menu options from $M$ whose price lies in $[c \cdot 2^i, c \cdot 2^{i+1})$ for an odd integer $i$, and $M_2$ denote the remaining menu options (which lie in $[c \cdot 2^i, c \cdot 2^{i+1})$ for an even power of $i$). It is easy to see that $M_1$ is oddly-priced and $M_2$ is evenly-priced. Then for all $\vec{v}$, we must have $p^{M_1}(\vec{v}) + p^{M_2}(\vec{v}) \geq p^M(\vec{v})$. This is because $\vec{v}$'s favorite menu

option from $M$ appears in one of the two menus, and is necessarily $\vec{v}$'s favorite option on that menu (and they pay non-zero from the other menu). Taking an expectation with respect to $\vec{v}$ yields that $\text{Rev}(\mathcal{D}, M_1) + \text{Rev}(\mathcal{D}, M_2) \geq \text{Rev}(\mathcal{D}, M)$, completing the proof. $\qquad\square$

*Proof of Claim 18.* Recall that $(1 + \varepsilon) \cdot ||\vec{v}||_1 \geq ||\vec{x}_i||_1$ for all $\vec{v} \in B_i$. Therefore, if we set a price of $||\vec{x}_i||_1 / (1 + \varepsilon)$ for the grand bundle, every $\vec{v} \in B_i$ would choose to purchase the grand bundle. This immediately implies the claim, as: $\text{BRev}(\mathcal{D}) \geq \frac{||\vec{x}_i||_1}{1+\varepsilon} \cdot \Pr_{\vec{v} \sim \mathcal{D}}\left[||\vec{v}||_1 \geq \frac{||\vec{x}_i||_1}{1+\varepsilon}\right] \geq \frac{||\vec{x}_i||_1}{1+\varepsilon} \cdot \Pr_{\vec{v} \sim \mathcal{D}}[\vec{v} \in B_i]$. $\qquad\square$

*Proof of Claim 19.* Recall that $\text{gap}_i^{X,Q} := \min_{j<i}\{\vec{x}_i \cdot (\vec{q}_i - \vec{q}_j)\}$, and that $\vec{q}_i := \vec{q}^M(\vec{x}_i)$. For any fixed $j < i$, recall that because $M$ was a truthful mechanism, we must have:

$$\vec{x}_i \cdot \vec{q}^M(\vec{x}_i) - p^M(\vec{x}_i) \geq \vec{x}_i \cdot \vec{q}^M(\vec{x}_j) - p^M(\vec{x}_j)$$
$$\Rightarrow \vec{x}_i \cdot (\vec{q}_i - \vec{q}_j) \geq p^M(\vec{x}_i) - p^M(\vec{x}_j)$$
$$\Rightarrow \vec{x}_i \cdot (\vec{q}_i - \vec{q}_j) \geq p^M(\vec{x}_i)/2.$$

The first line is simply restating incentive compatibility. The second line is basic algebra, and substituting $\vec{q}_i := \vec{q}^M(\vec{x}_i)$. The third line invokes the fact that $p^M(\vec{x}_i) \geq 2^{2(i-1)+a}$, while $p^M(\vec{x}_j) < 2^{2(j-1)+a+1} \leq 2^{2(i-1)+a-1}$. $\qquad\square$

*Proof of Observation 21.* This follows immediately from weak Lagrangian duality. For a quick refresher on weak Lagrangian duality, observe that for any feasible solution to the LP defining $\text{AlignGap}'(X)$ we must have $\vec{x}_i \cdot (c_i\vec{x}_i - c_{i-1}\vec{x}_{i-1}) - \text{sgap}_i \geq 0$. Therefore, for any feasible solution to the original LP, that solution is also feasible for $\text{LagRel}_1(X)$, and the objective is only larger. Therefore, the optimal solution to $\text{LagRel}_1(X)$ must be at least as large as $\text{AlignGap}'(X)$. $\qquad\square$

*Proof of Observation 22.* For all $i$, $\max\{0, \text{sgap}_i\} - \text{sgap}_i \leq 0$. When $\text{sgap}_i = 0$, the maximum is achieved (and $\text{sgap}_i := 0$ is feasible). Substituting $\max\{0, \text{sgap}_i\} - \text{sgap}_i = 0$ for all $i$ concludes the proof. $\qquad\square$

*Proof of Proposition 27.* To ease notation throughout the proof, we'll use the notation $\text{gap}_{\ell,j}^{X,Q} := \text{gap}_i^{X,Q}$, where $\vec{x}_i := \vec{x}_{\ell,j}$ ($\vec{x}_i$ is the $j^{th}$ point on layer $\ell$). We will also use the notation $(\ell', j') < (\ell, j)$ if $\ell' < \ell$, or $\ell' = \ell$ and $j' < j$ (that is, if the $j'^{th}$ point in the $\ell'^{th}$ layer comes before the $j^{th}$ point in the $\ell^{th}$ layer). To understand $\text{gap}_{\ell,j}^{X,Q}$, we need to understand which point "sets the gap" for $\vec{x}_{\ell,j}$, that is, which $(\ell', j') := \arg\min_{(\ell',j')<(\ell,j)}\{(\vec{q}_{\ell,j} - \vec{q}_{\ell',j'}) \cdot \vec{x}_i\}$.

We first analyze which point sets the gap for $\vec{x}_{\ell,j}$ (for even $\ell$; for odd $\ell$ the gap is zero and we don't care which point sets it), and observe that it must either be $\vec{q}_{\ell,j-1}$ or $\vec{q}_{\ell-2,n_{\ell-2}-1}$ (that is, it must be the previous point in the same layer, or the final point in the previous even layer).

**Claim 30.** *For all $j$, and all even $\ell$, $\text{gap}_{\ell,j}^{X,Q} = \vec{x}_{\ell,j} \cdot \vec{q}_{\ell,j} - \max\{\vec{x}_{\ell,j} \cdot \vec{q}_{\ell-2,n_{\ell-2}-1}, \vec{x}_{\ell,j} \cdot \vec{q}_{\ell,j-1}\}$.*[11]

*Proof of Claim 30.* First, note that $\text{gap}_{\ell,j}^{X,Q} := \min_{(\ell',j')<(\ell,j)}\{\vec{x}_{\ell,j} \cdot (\vec{q}_{\ell,j} - \vec{q}_{\ell',j'})\} = \vec{x}_{\ell,j} \cdot \vec{q}_{\ell,j} - \max_{(\ell',j')<(\ell,j)}\{\vec{x}_{\ell,j} \cdot \vec{q}_{\ell',j'}\}$. To conclude the proof, simply observe that the first component of $\vec{q}_{\ell',j'}$ is monotone increasing in $\ell'$ (for fixed $j'$), and the second component is monotone increasing in $j'$ (for fixed $\ell'$). Moreover, the second component of $\vec{q}_{\ell',n_{\ell'}-1}$ is 1, and this is the maximum possible. Also, both components of $\vec{x}_{\ell,j}$ are non-negative, and therefore we conclude that $\vec{x}_{\ell,j} \cdot \vec{q}_{\ell-2,n_{\ell-2}-1} \geq \vec{x}_{\ell,j} \cdot \vec{q}_{\ell',j'}$ whenever $(\ell', j') \leq (\ell-2, n_{\ell-2}-1)$ (in fact, this extends even to $(\ell', j') \leq (\ell-1, n_{\ell-1}-1)$ as no new $\vec{q}$ are introduced in layer $\ell - 1$). Also, $\vec{x}_{\ell,j} \cdot \vec{q}_{\ell,j-1} \geq \vec{x}_{\ell,j} \cdot \vec{q}_{\ell,j'}$ whenever $j' \leq j - 1$. $\qquad\square$

Now that we know that the gap is set either by the last point in the previous layer, or the previous point in the current layer, we can nail down $\text{gap}_{\ell,j}^{X,Q}$ exactly.

**Lemma 31.** *For all even $\ell > 2$, and all $j \in [0, n_\ell - 1]$: $\text{gap}_{\ell,j}^{X,Q} \geq \delta_\ell \frac{\sin(\theta_\ell)}{\sin((j+1)\theta_\ell)}$.*

---

[11]For simplicity of notation, define $\vec{q}_{0,j} = \vec{0} = \vec{q}_{\ell,-1}$ for all $\ell, j$.

*Proof of Lemma 31.* To prove the lemma, we simply compute the inner product of $\vec{x}_{\ell,j}$ with the three relevant vectors $\vec{q}_{\ell,j}, \vec{q}_{\ell-2,n_{\ell-2}-1}, \vec{q}_{\ell,j-1}$. To this end, recall that:

$$\vec{q}_{\ell,j} = (z_\ell, 1 - \delta_\ell \cot((j+1)\theta_\ell)),$$
$$\vec{q}_{\ell,j-1} = (z_\ell, 1 - \delta_\ell \cot(j\theta_\ell)),$$
$$\vec{q}_{\ell-2,n_{\ell-2}-1} = (z_{\ell-2}, 1).$$

Therefore, observe that

$$
\begin{aligned}
\vec{x}_{\ell,j} \cdot (\vec{q}_{\ell,j} - \vec{q}_{\ell,j-1}) &= \sin(j\theta_\ell) \cdot \delta_\ell \cdot (\cot(j\theta_\ell) - \cot((j+1)\theta_\ell)) \\
&= \sin(j\theta_\ell) \cdot \delta_\ell \cdot \left( \frac{\cos(j\theta_\ell)}{\sin(j\theta_\ell)} - \frac{\cos((j+1)\theta_\ell)}{\sin((j+1)\theta_\ell)} \right) \\
&= \delta_\ell \cdot \frac{\cos(j\theta_\ell)\sin((j+1)\theta_\ell) - \sin(j\theta_\ell)\cos((j+1)\theta_\ell)}{\sin((j+1)\theta_\ell)} \\
&= \delta_\ell \cdot \frac{\sin(\theta_\ell)}{\sin((j+1)\theta_\ell)}.
\end{aligned}
$$

Similarly,

$$
\begin{aligned}
\vec{x}_{\ell,j} \cdot (\vec{q}_{\ell,j} - \vec{q}_{\ell-2,n_{\ell-2}-1}) &= (\delta_\ell + \delta_{\ell-1}) \cdot \cos(j\theta_\ell) - \delta_\ell \cot((j+1)\theta_\ell) \cdot \sin(j\theta_\ell) \\
&\geq \delta_\ell \cdot \cos(j\theta_\ell) - \delta_\ell \cot((j+1)\theta_\ell) \cdot \sin(j\theta_\ell) \\
&= \frac{\delta_\ell}{\sin((j+1)\theta_\ell)} \left(\sin((j+1)\theta_\ell)\cos(j\theta_\ell) - \sin(j\theta_\ell)\cos((j+1)\theta_\ell)\right) \\
&= \delta_\ell \frac{\sin(\theta_\ell)}{\sin((j+1)\theta_\ell)}.
\end{aligned}
$$

This means that no matter which point sets the gap (or if one of the points does not exist), the gap is at least $\delta_\ell \frac{\sin(\theta_\ell)}{\sin((j+1)\theta_\ell)}$. $\qquad\square$

Finally, we need to sum over each even layer.

**Corollary 32.** *For any even $\ell > 2$, $\sum_{j=0}^{n_\ell-1} \mathrm{gap}_{\ell,j}^{X,Q} \geq \delta_\ell \cdot \ln(n_\ell)/2$.*

*Proof of Corollary 32.* Consider the following sequence of calculations:

$$
\begin{aligned}
\sum_{j=0}^{n_\ell-1} \mathrm{gap}_{\ell,j}^{X,Q} &\geq \sum_{j=0}^{n_\ell-1} \delta_\ell \frac{\sin(\theta_\ell)}{\sin((j+1)\theta_\ell)} \\
&\geq \delta_\ell \cdot (\theta_\ell - \theta_\ell^3/6) \cdot \sum_{j=0}^{n_\ell-1} \frac{1}{(j+1)\theta_\ell} \\
&\geq \delta_\ell \cdot (1 - \theta_\ell^2/6) \cdot \ln(n_\ell) \\
&\geq \delta_\ell \cdot \ln(n_\ell)/2
\end{aligned}
$$

Above, the first line follows from Lemma 31. The second line uses the fact that $\theta_\ell - \theta_\ell^3/6 \leq \sin(\theta_\ell) \leq \theta_\ell$, because $\theta_\ell \in [0, \pi/2]$. The third line follows as the $n^{th}$ harmonic sum is at least $\ln(n)$. The final line follows as $\theta_\ell^2/6 = \pi^2/(24(n_\ell - 1)^2) \leq 1/2$. $\qquad\square$

And finally, we can wrap up the proof of the proposition. Here, we just need to recall that $\delta_\ell := \frac{1}{\alpha n_\ell} = \frac{1}{\alpha \ell \ln^2(\ell)}$. Therefore, we conclude that:

$$
\sum_{\ell \text{ even}} \sum_{j=0}^{n_\ell-1} \mathrm{gap}_{\ell,j}^{X,Q} \geq \sum_{\ell \text{ even}} \delta_\ell \cdot \ln(n_\ell)/2 = \sum_{\ell \text{ even}} \frac{1}{2\alpha\ell \ln(\ell)} = \infty. \qquad\square
$$

# C Proof of Corollary 11

We prove Corollary 11 by making use of Theorem 2 combined with the sequence $X$ from Section 5. The only task is to confirm that $\text{ARev}(D) < \infty$ for the resulting $\mathcal{D}$, which essentially requires that we execute and analyze the construction fully. Let us quickly review the [HN19] construction, given as input a sequence $X$:

- Let $B$ be a very large constant, to be defined later.
- Let $\vec{v}_i := B^{2^i} \cdot \vec{x}_i / ||\vec{x}_i||_1$ (for all $i$).
- Let $\mathcal{D}$ sample $\vec{v}_i$ with probability $1/B^{2^i}$ (for all $i$).
- Let $\mathcal{D}$ sample $\vec{0}$ with probability $1 - \sum_{i \geq 1} 1/B^{2^i}$.

[HN19] establishes that the above construction yields Theorem 2 (for sufficiently large $B$, as a function of $\varepsilon$). To complete the proof of Corollary 11, we just need to relate $\text{ARev}(D)$ for this construction to $\text{AlignGap}(X)$.

**Proposition 33.** *The construction above yields a $\mathcal{D}$ satisfying* $\text{ARev}(\mathcal{D}) \leq \text{AlignGap}(X) + 1/B$.

*Proof.* Consider any mechanism $M$. We show that $\text{AlignGap}(X) \geq \text{ARev}(\mathcal{D}, M) - 1/B$. To see this, consider the following choice of $C$:

- If $\vec{v}_i$ is parallel to $\vec{q}^M(\vec{v}_i)$, set $c_i := ||\vec{q}^M(\vec{v}_i)||_2 / ||\vec{x}_i||_2$.
- If $\vec{v}_i$ is not parallel to $\vec{q}^M(\vec{v}_i)$, set $c_i := 0$.

We now need to lower bound $\text{sgap}_i^{X,C}$, when $i$ satisfies the first bullet. Observe that, because $M$ is truthful, we must have, for all $j < i$:

$$\vec{v}_i \cdot \vec{q}^M(\vec{v}_i) - p^M(\vec{v}_i) \geq \vec{v}_i \cdot \vec{q}^M(\vec{v}_j) - p^M(\vec{v}_j)$$
$$\Rightarrow p^M(\vec{v}_i) \leq p^M(\vec{v}_j) + B^{2^i} \vec{x}_i \cdot (c_i \vec{x}_i - c_j \vec{x}_j)/||\vec{x}||_1$$
$$\Rightarrow p^M(\vec{v}_i) \leq 2B^{2^{i-1}} + B^{2^i} \text{sgap}_i^{X,C}/||\vec{x}_i||_1$$

Above, the first line follows from incentive compatibility. The second line follows as $\vec{q}^M(\vec{v}_i) = c_i \vec{x}_i$ for all $i$ in the first bullet, and either $\vec{q}^M(\vec{v}_j) = c_j \vec{x}_j$, or $c_j = 0$. The final line follows by taking $j := \arg\min_{j<i}\{\vec{v}_i \cdot (c_i \vec{v}_i - c_j \vec{v}_j)\}$, and by observing that $\vec{v}_j$ cannot possibly pay more than their value for the grand bundle.

We can then conclude that:

$$\text{ARev}(\mathcal{D}, M) \leq \sum_i (2B^{2^{i-1}} + B^{2^i} \text{sgap}_i^{X,C}/||\vec{x}_i||_1)/B^{2^i}$$
$$\leq \sum_i 2/B^{2^{i-1}} + \text{AlignGap}(X)$$
$$\leq \text{AlignGap}(X) + 1/B.$$

$\square$

Because we can take $B$ as large as we like, we can construct a $\mathcal{D}$ such that $\text{ARev}(\mathcal{D})$ is arbitrarily close to $\text{AlignGap}(X)$, while also maintaining that $\text{Rev}(\mathcal{D})$ is arbitrarily close to $\text{MenuGap}(X)$. Because Theorem 9 provides a construction $X$ such that $\text{MenuGap}(X)/\text{AlignGap}(X) = \infty$, the [HN19] construction, with sufficiently large $B$, yields a $\mathcal{D}$ with $\text{Rev}(\mathcal{D})/\text{ARev}(\mathcal{D}) = \infty$, completing the proof of Corollary 11.