# OpenReview forum: "On Infinite Separations Between Simple and Optimal Mechanisms"
_NeurIPS.cc/2022/Conference — NeurIPS 2022 Accept_

### Official Review · Reviewer_92Mc · 2022-07-05

**Rating:** 7
**Confidence:** 3
**Soundness:** 3 good
**Presentation:** 4 excellent
**Contribution:** 3 good

**Summary:**

This paper studies separation between the simple and optimal mechanisms in terms of revenue when selling $k$ heterogeneous items to a single buyer. An arguably simple scheme is to offer to sell either all the items together at a fixed price or nothing (hence treating all the items together as a grand bundle). Let us call the revenue achieved by this strategy the "bundle revenue", we denote its value by $BRev(\mathcal D)$, where $\mathcal D$ is the distribution of the valuations of the buyer for each item (valuations for different items can be correlated). However, there is no reason to believe that this would give the best possible revenue. I.e. there is no reason to believe that $Rev(\mathcal D)$ (the optimal revenue for distribution $\mathcal D$) and $BRev(\mathcal D)$ are equal, or even close.

Indeed, a recent work [HN19] exhibits instances where the multiplicative gap between $BRev(\mathcal D)$ and $Rev(\mathcal D)$ is infinite. To achieve this result, they identify a geometric quantity that is based on defining two sequences of $k$-dimensional vectors where $X=(\overrightarrow{x_i})$ with $\overrightarrow{x_i}\in \mathbb R_{\geq 0}^k$ and $Q=(\overrightarrow{q_i})$ with $\overrightarrow{q_i}\in [0,1]^k$. From these two families, one can define a quantity called MenuGap$(X,Q)$ that is intimately linked to the multiplicative gap $Rev(\mathcal D)/BRev(\mathcal D)$.

Specifically, [HN19] shows that for any $X,Q$ and any $\epsilon>0$, there exists a distribution $\mathcal D$ such that $Rev(\mathcal D)/BRev(\mathcal D)>(1-\epsilon) \cdot$MenuGap$(X,Q)$.

This result raises 2 natural questions:

1) Is MenuGap$(X,Q)$ the "right" quantity to get multiplicative gaps $Rev(\mathcal D)/BRev(\mathcal D)$?
2) Looking at the result of [HN19], one can observe that the vectors $\overrightarrow{x_i}\in X$ and $\overrightarrow{q_i}\in Q$ are colinear for all $i$. In fact, *all* constructions of some MenuGap$(X,Q)$ quantity in the past literature satisfy this property. Does there exist constructions such that MenuGap$(X,Q)$ is big but only of $Q$ and $X$ are *not* colinear?

The authors answer both these questions. For the first one, they show that for any distribution $\mathcal D$ there exists two families of vectors $X,Q$ (with $X$ that is included in the support of $\mathcal D$) such that MenuGap$(X,Q)>Rev(\mathcal D)/(9\cdot BRev(\mathcal D))$. That is, if we do not care about constant factors, the theorem of [HN19] is "complete" in the sense that every distribution that witnesses a specific gap can be obtained through this theorem.

For the second question, the authors answer positively. There exists a set $X$ where MenuGap$(X,Q)$ can be unbounded for a suitable $Q$ but MenuGap$(X,Q)<6$ for any $Q$ that is colinear to $X$. This result reveals that there are constructions that are fundamentally different from previous works.

**Questions:**

I did not see any typo, the paper is in general clearly written. But here are a few remarks:

- Is there an intuition as to how you constructed the non-aligned example? It would be nice to explain a bit more if there is any intuition.

- if space permits, I would have put the proof of Claims 18-19 in the main body of the paper.

**Strengths And Weaknesses:**

The strengths are (in my opinion) the following:

- The paper is well-written and clear given the quite complicated definitions.
- The results seem non-trivial and correct.
- The results seem interesting. I am not an expert in mechanism design but the gap between simple and complicated mechanisms seems important to understand (and in particular its relation with this geometric quantity). This paper gives results in that direction.

I see no obvious weaknesses.

---

> ### Author Response · Authors · 2022-07-29
> **Response to Reviewer 92Mc**
>
> Thank you for the thorough review and thoughtful question. Below we give a high-level idea of our process in designing this construction that may be useful.
>
> The first half of the proof (up to Proposition 26) is all about upper-bounding AlignGap(X). To this end, we make the following relaxations:
> - Certainly, the maximum gap between $x_i$ and any previous point is at least the maximum gap between $x_i$ and $x_{i-1}$. (Observation 21). This greatly simplifies the math.
> - Claim 24 turns this observation into a bound, for any sequence that places consecutive points on “shells” with the same $\ell_2$ norm. The idea to use “shells” appears in [Hart and Nisan 2019].
> - The precise choice of the number of points per shell, and the $\ell_2$ norm of each shell, is chosen specifically to bound the expression in Proposition.
> - So to wrap up the first half, we knew that we wanted to have a construction based on consecutive shells, and we wanted a mathematically clean way to upper bound MenuGap(X). The precise choices we made allow this approach to work.
>
> The second half of the proof is about lower-bounding MenuGap(X). To this end:
> - We knew that we wanted to pick Q so that the “best” $q_j$ for a point $x_i$ corresponds to a point near it, and we specifically chose Q to satisfy this property.
> - Then, we face a (not particularly clean) optimization problem subject to these constraints. We played around with some small examples and learned that good solutions increased the y-coordinate within a shell (keeping the x-coordinate fixed), and increased the x-coordinate only between shells. Restricting to this structure also made it cleaner to write the constraints from the previous bullet.
> - Then, it is just a matter of understanding what conditions are needed in order for the sum of resulting gaps to diverge.
> - There’s actually not a lot of wiggle-room here: note that our infinite sum for AlignGap sums to $\frac{1}{\ell \ln^2 (\ell)}$, which converges, while our infinite sum for MenuGap sums to $\frac{1}{\ell \ln(\ell)}$, which diverges. Because there’s not much wiggle room, it’s hard to give a clear, strong intuition for why everything works out, but we hope that the above notes at least help the reviewer understand how we approached it.
>
> As a final note, this was a very tough construction for us to develop, with a lot of trial and error. This final version is also the outcome of a series of simplifications after our first correct construction/proof of this result. We suspect that there isn’t a significantly cleaner construction with strong intuition, and some amount of care and subtlety is necessary to get such a result.

---

> > ### Comment · Reviewer_92Mc · 2022-08-08
> > **Thanks for the response!**
> >
> > Thank you for this nice answer. Indeed, it seems that it was non trivial to come up with this construction. After reading your response to my review and other reviews, I still think this is a good paper and maintain my score.

---

### Official Review · Reviewer_aofd · 2022-07-11

**Rating:** 6
**Confidence:** 3
**Soundness:** 4 excellent
**Presentation:** 4 excellent
**Contribution:** 3 good

**Summary:**

The authors study Bayesian mechanism design where the goal is to maximize expected revenue of selling k (heterogeneous) items to a single buyer with additive valuations.
The authors investigate when the optimal revenue is infinite while the best-possible revenue from selling all items in a grand bundle is finite; this question has significance for understanding the power of any mechanism (such as deterministic bundle pricing) that has a finite menu size.
A reduction from previous work provides a recipe for constructing such "infinitely-separated" instances, by constructing instances with infinite MenuGap.

The authors' first main result is that infinite MenuGap is also a _necessary_ condition for an instance to be infinitely-separated.
A subclass of such infinite-separated instances can be characterized by those also with an infinite SupGap, which is always less than MenuGap.
Previous constructions of infinite MenuGap always involved infinite SupGap, suggesting that infinite SupGap may also be necessary for an instance to be infinitely-separated.
However, the authors' second main result is to show that this is false---in fact, they construct an infinitely-separated instance that has not only finite SupGap, but even finite AlignGap, a concept introduced by the authors.

**Questions:**

I have zero clarification questions because the manuscript was crystal clear.

If the authors have space, perhaps they could entertain the one weakness I pointed out.  I find the questions being studied centered about pathological worst-case constructions, and would love to know whether these instances are believed to arise in practice, or whether answering these questions is useful for informing auction design in practice.

**Strengths And Weaknesses:**

+: This paper has exceptional writing: concise, yet precise and an easy to follow train of thought.

+: The results are quite complete and provide a comprehensive understanding of when Rev/BRev = infty.

+: I agree with the authors that AlignGap is the right concept to use to make their second main result stronger, and find this to be generally useful.

-: Although I like the theory, I'm not sure why I should care about characterizing the pathological instances with Rev/BRev=infty, especially in a broader Neurips context.  The paper also does not make much effort to address this, until the end of the Conclusion, and even those connections are used to answer further mostly-theory questions.  Anyways, this discussion could be outside the scope of this particular paper, and I am overall quite positive about it.

---

> ### Author Response · Authors · 2022-07-29
> **Response to Reviewer aofd**
>
> Thank you for the thorough review and thoughtful question. In terms of relevance to practice (also see our response to reviewer qmDH), we’d argue that the field of theoretical mechanism design as a whole has a significant impact on practice. E.g. Ad Auctions, FCC Spectrum auctions, and many other modern CS+Econ auction theory results genuinely influence design choices made in practice. And we’d argue that the field as a whole can have greater impact on practice with a better understanding of the limits of theoretical tractability.
>
> For your specific question about whether these instances are believed to arise in practice, instances where simple auctions are very suboptimal are likely to arise in practice. But, of course, almost certainly the gap is nowhere near infinite. It is an active (but nascent) research area to understand what distributional assumptions imply some form of tractability, and whether these assumptions make sense in practice (see our response to reviewer qmDH on one recent paper using topic models). Our work has direct impact on this research agenda, which aims to understand classes of tractable distributions that arise in practice.

---

### Official Review · Reviewer_nmG2 · 2022-07-11

**Rating:** 7
**Confidence:** 4
**Soundness:** 4 excellent
**Presentation:** 3 good
**Contribution:** 3 good

**Summary:**

The authors focus on multi-item mechanism design with valuations distributions that are correlated across items. They study conditions under which there is an infinite gap in revenue between the optimal mechanism and grand bundling/simple mechanisms.

**Questions:**

- Do the authors of the paper have a sense of whether it is possible to construct/how, for any distribution D with infinite gap, some set X such that MenuGap(X) is also infinite? I think saying a word about this would tighten the results of the paper

**Strengths And Weaknesses:**

I really enjoyed reading this paper. I think it provides a useful understanding of when optimal mechanisms can have finite vs infinite size for multi-item auctions.
- The topic is important and interesting, as it sheds light on when simple and practical mechanisms perform well, in non-trivial settings in which the valuation distribution can be correlated across items. -  - Theorem 6 shows that the framework of Hart and Nisan is essentially "tight". Hart and Nisan showed the following result: let M be a mechanism (of infinite menu size). If for some distribution D, M gets infinitely more revenue than any mechanism with a finite menu) then MenuGap(X) is infinite where X is related to the support of D.
The current paper shows the converse, showing that this characterization is essentially tight: for any distribution D with an infinite gap, there is a set X of points in the support of D such that MenuGap(X) is infinite.
In turn, the paper shows that the quantity MenuGap is essentially the right metric to characterize whether the gap between any finite and the optimal mechanism is infinite in terms of revenue.
- Despite Theorem 6 stating the existence of an X as described above, the proof of Theorem 6 actually shows how to construct this X if we know the distribution D. This provides a way to verify systematically/algorithmically when the revenue gap is finite (it suffices to construct this X and compute its MenuGap. If it’s finite, then the revenue gap is finite).
- The authors extend their results to show that AlignGap, a natural but more restrictive version of MenuGap motivated from previous work does not enjoy the same property: there can be an infinite gap between AlignGap and MenuGap. This means that there exists situations where AlignGap(X) is finite, but there exists a distribution D on (a rescaling of) support X with an infinite revenue gap.
- I believe the techniques are non-trivial and interesting (for example the proof of Theorem 6 requires throwing away menu options that have low prices, and dividing the support of the valuation distribution according to price intervals of growing size), yet the paper is written in such a way that they are very easy to follow.


Weakness:
- I believe the main weakness of the paper is that even though the paper shows that MenuGap seems to be the right/tight measure of complexity of the problem, there is a slight mismatch between theorem 2 and 6. In particular, Theorem 2 does not allow you to argue directly whether the ratio of revenues is infinite for a given distribution D, as it does not tell you how to find a X’ such that MenuGap(X’) lower bounds the ratio of revenues between optimal and simple mechanisms. Rather, it constructs D from X’. It would be useful to provide such a construction for the characterization to be fully tight algorithmically.

Despite this weakness, I think this is a good paper that provides additional understanding on what optimal mechanisms look like in multi-dimensional mechanism design with correlated valuations, and I would like to see the paper accepted.

---

> ### Author Response · Authors · 2022-07-29
> **Response to Reviewer nmG2**
>
> Thank you for the thorough review, and also for the thoughtful question. Here is what we know about constructing sequences that are subsets of the support of D with a large menu gap:
>
> 1) As you note in your review, the proof of Theorem 6 is constructive, given an IC mechanism M for D. That is, given a mechanism M with $Rev^M(D) = x$, the proof of Theorem 6 produces a sequence X in the support of D with $MenuGap(X) \geq x/9$.
>
> 2) If $Rev(D) = \infty$, this constructive approach for producing a sequence X would be to: a) find a mechanism M such that $Rev^M(D) = \infty$, and then b) go through the same construction in the proof of Theorem 6 (with minor notational adjustments).
>
> 3) If (2) does not feel “constructive enough,” since it requires finding/describing an infinite object, an alternative approach would be to find finite sequences with arbitrarily large MenuGap. This can be done by a) truncating the support of D by zeroing out any valuation vectors with any value outside of $[\varepsilon, 1/\varepsilon]$, b) computing an approximately-optimal auction for this truncated distribution via discretization + linear programming, c) run the constructive procedure in the proof of Theorem 6. As $\varepsilon \rightarrow 0$, the resulting sequence will have MenuGap going to $\infty$.

---

> > ### Comment · Reviewer_nmG2 · 2022-08-07
> > **Thanks for the response!**
> >
> > Thanks a ton! I think it'd be nice to incorporate this discussion somewhere in the paper maybe as a remark. Overall, while I have not changed my score, this reinforces my opinion that this is a strong paper that I want to see accepted at NeurIPS.

---

### Official Review · Reviewer_qmDH · 2022-07-12

**Rating:** 3
**Confidence:** 2
**Soundness:** 3 good
**Presentation:** 2 fair
**Contribution:** 1 poor

**Summary:**

This paper considers the problem of 1 buyer, 1 seller selling k items. In the context of this problem, they look at a mechanism from prior work that relates the ratio of optimal revenue to the myerson's optimal revenue for the total bundle and the menuGap. In particular, prior work showed that there exists a sequence generated using a framework such that ratio of optimal revenue to the myerson's optimal revenue via showing menu gap is infinity. The first result in this paper is that this framework is without loss of generality. The second result in this paper is that there exists a sequence where align gap is small but the menu gap is infinity.

**Questions:**

Could you please elaborate why the results obtained in this paper are significant/relevant for the larger machine learning community? In my opinion, I think the paper needs to be written more broadly and accessible

**Limitations:**

This is a primarily theory paper. So there aren't much potential negative societal impact.

**Strengths And Weaknesses:**

Originality

+ This paper is a primarily theoretical/conceptual paper where the goal is to understand prior work on revenue maximizing schemes for selling multiple items. This paper is original in that it pushes the state of understanding on this problem and some of the constructed schemes.

Quality

+ The obtained results, the corresponding proofs are of high-quality. The results are non-trivial to obtain and definitely improves the understanding of this specific problem.

Clarity

+ Overall the paper is well-written however, the target audience is a small set of people who understand the theory of prior work. The paper does not make much efforts to make this broadly accesible and/or mention why the results are significant.

Significance

This is the primary qualm I have with this submission. The paper is extremely niche and narrow to be of interest to a large machine learning community (e.g., Neurips). The paper seems to be making some incremental understanding over prior theorems. It is very hard for me to see why the results presented in this paper are important or significant. I believe this paper may be suitable for a niche conference for the appropriate community and don't believe this is appropriate for a conference like neurips.

---

> ### Author Response · Authors · 2022-07-29
> **Response to Reviewer qmDH**
>
> Thank you for the thoughtful review and question. We will elaborate more on the context, significance, and relevance to the ML community in the final version of the paper. Here is some further context that may help.
>
> First, the [Briest, Chawla, Kleinberg, Weinberg 2010/2015, Hart and Nisan 2013/2019] infinite-gap construction has dominated the study of approximately-optimal multi-dimensional auctions since essentially the beginning of the field (only [Chawla, Hartline, Kleinberg 2007] predates these). These results establish that non-trivial approximation guarantees, of any form, are impossible for arbitrarily correlated distributions. For the past decade, these two results justified the assumption of "independent items," and simple/approximately optimal multi-item auctions with "independent items" has been among the most dominant agendas at the intersection of CS+Econ for the past decade. So, we'd argue that a deep understanding of these constructions is of high significance because these constructions themselves have had a substantial direct impact in guiding a dominant research agenda for the past decade.
>
> Second, now that the agenda of simple/approximately optimal multi-item auctions with "independent items" has matured, very recent work tries to extend these results beyond the "independent items" regime. The manner in which this is done draws on tools relevant and familiar to the ML community. For example, very recent work of [Cai and Daskalakis 2022] uses distributional assumptions similar to topic models, and designs approximately-optimal auctions in this setting. In this context, our work helps clarify the limits of such approaches and models: a model is tractable if and only if it steers clear of the [Hart and Nisan 2019] framework. Beyond this specific direction, we note that there is a significant community at the intersection of machine learning and multi-item auction design, and understanding the limits of tractability is important for this community.

---

### Author Response · Authors · 2022-07-29
**Response to all reviewers**

We would like to thank all reviewers for their constructive feedback. We will incorporate the valuable suggestions from all reviewers in the final version of this paper. Below we respond to each reviewer individually.

---

### Meta-Review · Area_Chair_u2AS · 2022-08-27

**Recommendation:** Accept
**Confidence:** Certain

**Metareview:**

This very well-written studies Bayesian mechanism design where we aim to maximize the expected revenue of selling k items to a single buyer with additive valuations. The authors investigate when the optimal revenue is infinite while the best-possible revenue from selling all items in a grand bundle is finite; this question has significance for understanding the power of any mechanism---such as deterministic bundle pricing---that has a finite menu size. A reduction from previous work provides a recipe for constructing such "infinitely-separated" instances, by constructing instances with infinite "MenuGap". The first main result is that infinite MenuGap is also a necessary condition for an instance to be infinitely-separated. A subclass of such infinite-separated instances can be characterized by those also with an infinite "SupGap"---a gap that is upper-bounded by the MenuGap. Previous constructions of infinite MenuGap always involved infinite SupGap, suggesting that infinite SupGap may also be necessary for an instance to be infinitely-separated; the second main result falsifies this by an infinitely-separated instance that has finite SupGap.

Despite some concerns about fit with NeurIPS, I find the overall fit quite good.


**Award:**

No

---

### Decision · Program_Chairs · 2022-09-14

Accept